# Particle representation for $NN\bar{K}$ system

**Branislav Vlahovic[1], Igor Filikhin[1⋆] and Roman Ya. Kezarashvili[2]**

**1** North Carolina Central University, Durham, NC 27707, USA
**2** The City University of New York, Brooklyn, NY 11201, USA

⋆ ifilikhin@nccu.edu

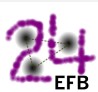 *Proceedings for the 24th edition of European Few Body Conference,
Surrey, UK, 2-6 September 2019

## Abstract

**In the framework of the Faddeev equations in configuration space we perform an analysis of quasi-bound state of the $NN\bar{K}$ system within a particle model. In our approach, the system $NN\bar{K}(s_{NN} = 0)$ ($NN\bar{K}(s_{NN} = 1)$) is described as a superposition of $ppK^-$ and $pn\bar{K}^0$ ($nn\bar{K}^0$ and $pnK^-$) states, which is possible due to a particle transition. The relation of the particle model to the theory of a two-state quantum system is addressed and discussed taking into account the possibilities of deep and shallow $NN\bar{K}(s_{NN} = 0)$ quasi-bound states.**

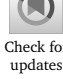
## 1 Introduction

Over 50 years ago a study of three possible isospin configurations of the $\bar{K}NN$ system led Nogami [1] to the assumption of the possible existence of the bound state in this system: in a antikaon-nucleons system the presence of the $K^-$ meson attracts two unbound protons to form a $K^-pp$ cluster. Calculations performed by Akaishi and Yamazaki [2] have predicted the possible existence of discrete nuclear bound states of $\bar{K}$ in few-body nuclear systems and this prediction was confirmed by several subsequent publications (see [3] and references herein).

The theoretical prediction of the kaonic nuclei stimulated the experimental search of the deeply bound states of $\bar{K}$ in few-body nuclear clusters through different nuclear processes. Both experimental and theoretical advances have been made in the last two decade for study kaonic nuclear $K^-pp$ state. The experimental and theoretical status of the $K^-pp$ is summarized in Refs. [4–8].

Based on the results of experimental search of the $K^-pp$ cluster, one can conclude that the situation is still controversial and the existence of the $K^-pp$ bound state has not yet been established [4]. However, the most recently, the J-PARC E15 collaboration reported the observation of a distinct peak in the $\Lambda p$ invariant mass spectrum of ${}^3\text{He}(K^-, \Lambda p)n$, well below $m_K + 2m_p$, *i.e.*, the mass threshold of the $K^-$ meson to be bound to two protons. The simplest

fit to the observed peak gives a Breit–Wigner pole position at $47\pm3(stat.)^{+3}_{-6}(sys.)$ MeV having a width $115 \pm 7(stat.)^{+10}_{-20}(sys.)$ MeV, which the authors claim as a new form of the nuclear bound system with strangeness $- K^-pp$ [9]. At the same time, one can note the theoretical analysis of this experiment in Ref. [8], where the two quasi-bound states was predicted. The deeply bound state has energy about -100 MeV and second one has energy about -50 MeV.

The kaonic strange dibaryon $\bar{K}NN$ represent a three-body systems and theoretically have been treated in the framework of a few-body physics approaches: the variational method including the framework of antisymmetrized molecular dynamics, method of Faddeev equations in momentum and configuration representations, the Faddeev equations in the fixed center approximation and method of hyperspherical harmonics in configuration space and momentum representation (see reviews [5–8] and references herein). Calculations for a binding energy and width of the kaonic three-body system are performed using different potentials for the $NN$ interaction, as well as different potentials for the description of the kaon–nucleon interaction. The latter are the energy-independent phenomenological $\bar{K}N$ potential and the energy-dependent chiral $\bar{K}N$ interaction. All aforementioned approaches predict the existence of a bound state for the $K^-pp$. The $K^-pp$ cluster binding energy was theoretically estimated to be approximately 10–20 MeV for energy-dependent chiral interactions and 40–90 MeV for energy-independent $\bar{K}N$ interactions. The predicted values for the binding energy and the width are in considerable disagreement: 9–95 MeV and 20–110 MeV, respectively. Interestingly enough that theoretical models have disagreements related to theoretical values of the binding energy and decay width and have a large ambiguity depending on the $\bar{K}N$ interaction models and the calculation methods. There are systematic discrepancies between the theoretical predictions and experimental observations. For the theoretical status of $K^-pp$ refer to [5–7] and references therein.

In the present work we study the quasi-bound state of the system $NN\bar{K}$ within the method of Faddeev equations in configuration space. We are considering the system $NN\bar{K}(s_{NN} = 0)$ as a two-state quantum system $ppK^-/np\bar{K}^0$ within a particle representation. The latter allows us to analyze the quasi-bound state of the system $NN\bar{K}$ within two-level approach by considering it as a superposition of $K^-pp/\bar{K}^0pn$ in the framework of the potential model using $NN$ potential and energy-independent effective $\bar{K}N$ interactions.

## 2 Particle representation for $NN\bar{K}(s_{NN} = 0)$ system: $ppK^-$ and $np\bar{K}^0$ channels

In the presented work we restrict the model space to the $s$-wave approach. The Coulomb interaction is not taken into account and the mass differences for the $\bar{K}^0$ and $K^-$ mesons (5.1 MeV [10]), and neutron and proton (1.3 MeV [11]) are ignored. This input is corresponding to one used within isospin formalism consideration. In this scenario isospin singlet $N\bar{K}$ potential is the same for $K^-p$ and $\bar{K}^0n$ interactions. Therefore, for $NN\bar{K}(s_{NN} = 0)$ system, the singlet isospin configurations $(K^-p)p$ and $(\bar{K}^0n)p$ have to be equivalent. However, the $pK^-$ and $n\bar{K}^0$ systems are different when are taken into account the presence of the Coulomb force in $pK^-$ system and mass differences. In Fig. 1 are presented the results for calculation for the mass differences for the systems $n + \bar{K}^0$ and $p + K^-$ using Akaishi and Yamasuki (AY) potential [12], when the Coulomb repulsion between the proton and $K^-$ is ignored. These systems are separated by the difference of the masses with the gap of about 5 MeV as is shown in Fig. 1. This value is small with respect to the total mass of each system, however, it is significant in the energy scale relatively to the binding energy 30 MeV of the $N\bar{K}$.

The symmetry of the isospin picture which one wants to describe in the terms for $ppK^-$ and $np\bar{K}^0$ channels is violated in the $np\bar{K}^0$ system. The system $ppK^-$ is described using two

potentials ($v_{pp}$ and $v_{pK-}$), while for the description of the system $np\bar{K}^0$ we are using three different potentials: $v_{np}$, $v_{p\bar{K}^0}$ and $v_{n\bar{K}^0}$. The latter two are related to the interaction of $\bar{K}^0$ with the proton and neutron, respectively. Following Ref. [13] we are considering the system $NN\bar{K}(s_{NN} = 0)$ using the "particle representation" instead of the isospin formalism.

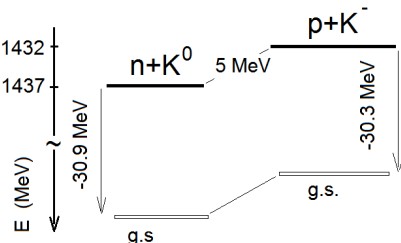

Figure 1: The mass differences for the systems $n + \bar{K}^0$ and $p + K^-$. The binding energies of the ground states for the $n + \bar{K}^0$ and $p + K^-$ are calculated with the AY effective isospin singlet $N\bar{K}$ potential [12]. The Coulomb force is not included in calculations for $p + K^-$ system.

In the present work, we consider the particle picture for the $NN\bar{K}$ system and propose to describe it as a two-level system $ppK^-/np\bar{K}^0$ by mixing $ppK^-$ and $np\bar{K}^0$ configurations. The two-level also known as two-state system is a quantum system that can exist in any quantum superposition of two independent and physically distinguishable quantum states [14]. We assume that in the systems $ppK^-$ and $np\bar{K}^0$, the $n\bar{K}^0$ and $pK^-$ interactions are equivalent to singlet isospin $N\bar{K}$ interaction, and the $p\bar{K}^0$ interaction equals to the triplet $N\bar{K}$ isospin interaction.

In the particle representation, the wave function $\Psi$ of the coupled $ppK^-/np\bar{K}^0$ system is a column vector. Let's decompose the wave functions of the systems $ppK^-$ and $np\bar{K}^0$ into the usual Faddeev components [15] $\psi_{ppK^-} = U_1 + W_1 + Y_1$ and $\psi_{np\bar{K}^0} = U_2 + W_2 + Y_2$, respectively. The wave function $\Psi$ of the coupled $ppK^-/np\bar{K}^0$ system is presented as a superposition of the wave functions of each system as:

$$\Psi = A\psi, \qquad \psi = \begin{pmatrix} \psi_1 \\ \psi_2 \end{pmatrix} = \begin{pmatrix} \psi_{ppK^-} \\ \psi_{pn\bar{K}^0} \end{pmatrix} = \begin{pmatrix} U_1 \\ U_2 \end{pmatrix} + \begin{pmatrix} W_1 \\ W_2 \end{pmatrix} + \begin{pmatrix} Y_1 \\ Y_2 \end{pmatrix}, \qquad (1)$$

$$A = \begin{pmatrix} \alpha & -\beta \\ \beta & \alpha \end{pmatrix}, \qquad \alpha = \sin\theta, \qquad \beta = \cos\theta. \qquad (2)$$

Here the parameter $\theta$ defines the coupling strength and $A$ is an unitary matrix: $AA^T = I$.

The Schrödinger equation for the $ppK^-$ and $np\bar{K}^0$ systems can be written in the following matrix form:

$$(H_0 + V_{pp,\,np} + V_{pK^-,\,p\bar{K}^0} + V_{pK^-,\,n\bar{K}^0} - E)\psi = 0. \qquad (3)$$

In Eq. (3) $H_0$ is the kinetic energy operator of three particles and the interactions are defined as

$$V_{pp,\,pn} = \begin{pmatrix} v_{pp} & 0 \\ 0 & v_{pn} \end{pmatrix}, V_{pK^-,\,p\bar{K}^0} = \begin{pmatrix} v_{pK^-} & 0 \\ 0 & v_{p\bar{K}^0} \end{pmatrix}, V_{pK^-,\,n\bar{K}^0} = \begin{pmatrix} v_{pK^-} & 0 \\ 0 & v_{n\bar{K}^0} \end{pmatrix},$$

where the $v_{pp}$ and $v_{np}$ are spin singlet components of the $NN$ potential. Following Ref. [13] the $v_{pK^-}$ and $v_{n\bar{K}^0}$ potentials are chosen as isospin singlet, while $v_{p\bar{K}^0}$ isospin triplet components of the $N\bar{K}$ potential, respectively. The isospin singlet $N\bar{K}$ state is related to the deeply quasi-bound state $\Lambda(1405)$ which is represented as the $pK^-$ system (see Fig. 1).

Let us apply the unitary transformation $A$ to (3) taking into account that $AA^T = I$ and the first three terms in Eq. (3) are invariants relative to the $A$ transformation due to the simplification: $m_p = m_n$, $m_{K^-} = m_{\bar{K}^0}$, $v_{pp} = v_{pn}$, $v_{pK^-} = v_{n\bar{K}^0}$. As a result we obtain the following equation for the wave function $\Psi$ of the coupled $ppK^-/np\bar{K}^0$:

$$((H_0 + V_{pp,np} + V_{pK^-,n\bar{K}^0})I + A(V_{pK^-,p\bar{K}^0})A^T - E)\Psi = 0. \tag{4}$$

The potential $V_{pK^-,p\bar{K}^0}$ generates the matrix $A(V_{pK^-,p\bar{K}^0})A^T$:

$$V = \begin{pmatrix} V_{11} & V_{12} \\ V_{21} & V_{22} \end{pmatrix}, \tag{5}$$

where $V_{11} = \alpha^2 v_{pK^-} + \beta^2 v_{p\bar{K}^0}$, $V_{12} = V_{21} = \alpha\beta(v_{pK^-} - v_{p\bar{K}^0})$, $V_{22} = \beta^2 v_{pK^-} + \alpha^2 v_{p\bar{K}^0}$. Thus, the matrix $A$ describes the coupling $ppK^-/np\bar{K}^0$ via non-diagonal elements $V_{12}$ and $V_{21}$. The isospin states of $ppK^-$ and $np\bar{K}^0$ are the same and the possible transformation $pK^- \to n\bar{K}^0$ can be described as a coupling between the systems. One should note that non-diagonal elements $V_{12}$ and $V_{21}$ of the matrix (5) make the channel coupling. The coefficients $\alpha$ and $\beta$ can be chosen to present a coupling between the channels.

Now the Schrödinger equation (4) for the coupled $ppK^-/np\bar{K}^0$ system can be written for the corresponding Faddeev components. These components satisfy the following deferential Faddeev equations (DFE):

$$\begin{aligned} (H_0^U + v_{pp} - E)U_1 &= -v_{pp}(W_1 + Y_1), \\ (H_0^W + v_{pK^-} - E)W_1 &= -v_{pK^-}(U_1 + Y_1), \\ (H_0^Y + V_{11} - E)Y_1 + V_{12}Y_2 &= -V_{11}(U_1 + W_1) - V_{12}(U_2 + W_2), \\ (H_0^U + v_{pn} - E)U_2 &= -v_{pn}(W_2 + Y_2), \\ (H_0^W + v_{n\bar{K}^0} - E)W_2 &= -v_{n\bar{K}^0}(U_2 + Y_2), \\ (H_0^Y + V_{22} - E)Y_2 + V_{21}Y_1 &= -V_{22}(U_2 + W_2) - V_{21}(U_1 + W_1). \end{aligned} \tag{6}$$

If $\alpha = 1$, $\beta = 0$ the system of equations (6) decouples and the first three equations describe the kaonic cluster $ppK^-$, while last three equations describe the kaonic cluster $pn\bar{K}^0$. In the case of a weak channel coupling one can assume that $\alpha \approx 1$ and $\beta \approx 0$. Thus, describing $NN\bar{K}$ ($s_{NN} = 0$) system we consider two separate states $ppK^-$ and $pn\bar{K}^0$. For the first one, the $s$-wave approach based on the DFE leads to the following equations [16]:

$$\begin{aligned} (H_0^U + v_{pp} - E)\mathcal{U} &= -v_{pp}(\mathcal{W} + \mathcal{Y}), \\ (H_0^W + v_{pK^-} - E)\mathcal{W} &= -v_{pK^-}(\mathcal{U} + \mathcal{Y}), \\ (H_0^Y + v_{pK^-} - E)\mathcal{Y} &= -v_{pK^-}(\mathcal{U} + \mathcal{W}), \end{aligned} \tag{7}$$

where $v_{pK^-}$ corresponds to the isospin singlet potential of the $N\bar{K}$ interaction. The $s$-wave approach for the $np\bar{K}^0$ system leads to the following DFE:

$$\begin{aligned} (H_0^U + v_{np} - E)\mathcal{U} &= -v_{np}(\mathcal{W} + \mathcal{Y}), \\ (H_0^W + v_{n\bar{K}^0} - E)\mathcal{W} &= -v_{n\bar{K}^0}(\mathcal{U} + \mathcal{Y}), \\ (H_0^Y + v_{p\bar{K}^0} - E)\mathcal{Y} &= -v_{p\bar{K}^0}(\mathcal{U} + \mathcal{W}). \end{aligned} \tag{8}$$

In Eq. (8) $v_{p\bar{K}^0}$ and $v_{n\bar{K}^0}$ are chosen to be correspond to isospin singlet and triplet components of the $N\bar{K}$ potential, respectively.

A particular interest presents the case when $\alpha = \beta = \frac{1}{\sqrt{2}}$, which means strong coupling between the channels, and therefore we have $v^+ \equiv V_{11} = V_{22} = \frac{v_{pK^-} + v_{p\bar{K}^0}}{2}$ and

$v^- \equiv V_{12} = V_{21} = \frac{v_{pK^-} - v_{p\bar{K}^0}}{2}$ and the system (6) becomes:

$$
\begin{aligned}
(H_0^U + v_{pp} - E)U_1 &= -v_{pp}(W_1 + Y_1) \ , \\
(H_0^W + v_{pK^-} - E)W_1 &= -v_{pK^-}(U_1 + Y_1) \ , \\
(H_0^Y + v^+ - E)Y_1 + v^- Y_2 &= -v^+(U_1 + W_1) - v^-(U_2 + W_2) \ , \\
(H_0^U + v_{pn} - E)U_2 &= -v_{pn}(W_2 + Y_2) \ , \\
(H_0^W + v_{n\bar{K}^0} - E)W_2 &= -v_{n\bar{K}^0}(U_2 + Y_2) \ , \\
(H_0^Y + v^+ - E)Y_2 + v^- Y_1 &= -v^+(U_2 + W_2) - v^-(U_1 + W_1) \ .
\end{aligned}
\tag{9}
$$

One can use the unitary transformation $B = \frac{1}{\sqrt{2}}\begin{pmatrix} 1 & 1 \\ -1 & 1 \end{pmatrix}$ for $Y_1$ and $Y_2$ in the following form: $Y' = BY$ or $Y = B^T Y'$, where $Y = (Y_1, Y_2)^T$ and

$$
B\begin{pmatrix} v^+ & v^- \\ v^- & v^+ \end{pmatrix}B^T = \begin{pmatrix} v_{pK^-} & 0 \\ 0 & v_{pK^0} \end{pmatrix}
$$

to simplify the set of Eqs. (9). The transformation leads to the following set of equations:

$$
\begin{aligned}
(H_0^U + v_{pp} - E)U_1 &= -v_{pp}(W_1 + \tfrac{1}{2}(Y_1 - Y_2)) \ , \\
(H_0^W + v_{pK^-} - E)W_1 &= -v_{pK^-}(U_1 + \tfrac{1}{2}(Y_1 - Y_2)) \ , \\
(H_0^Y + v_{pK^-} - E)Y_1 &= -v_{pK^-}(U_1 + W_1 + U_2 + W_2) \ , \\
(H_0^U + v_{pn} - E)U_2 &= -v_{pn}(W_2 + \tfrac{1}{2}(Y_1 + Y_2)) \ , \\
(H_0^W + v_{n\bar{K}^0} - E)W_2 &= -v_{n\bar{K}^0}(U_2 + \tfrac{1}{2}(Y_1 + Y_2)) \ , \\
(H_0^Y + v_{p\bar{K}^0} - E)Y_2 &= -v_{p\bar{K}^0}(U_2 - U_1 + W_2 - W_1) \ .
\end{aligned}
\tag{10}
$$

The set of Eqs. (10) allows us to make comparison with Eqs. (7) - (8) which describe the independent $ppK^-$ and $np\bar{K}^0$ systems. Taking into account that $v_{pp} = v_{pn}$ and $v_{pK^-} = v_{n\bar{K}^0}$, the set (10) can be rewritten as

$$
\begin{aligned}
(H_0^U + v_{pp} - E)A_1 &= -v_{pp}(B_1 + Y_1) \ , \\
(H_0^W + v_{pK^-} - E)B_1 &= -v_{pK^-}(A_1 + Y_1) \ , \\
(H_0^Y + v_{pK^-} - E)Y_1 &= -v_{pK^-}(A_1 + B_1) \ , \\
(H_0^U + v_{pn} - E)A_2 &= -v_{pn}(B_2 + Y_2) \ , \\
(H_0^W + v_{n\bar{K}^0} - E)B_2 &= -v_{n\bar{K}^0}(A_2 + Y_2) \ , \\
(H_0^Y + v_{p\bar{K}^0} - E)Y_2 &= -v_{p\bar{K}^0}(A_2 + B_2) \ ,
\end{aligned}
\tag{11}
$$

where $A_1 = U_1 + U_2$, $A_2 = U_2 - U_1$, $B_1 = W_1 + W_2$, $B_2 = W_2 - W_1$. As a result, one can see that the set of Eqs. (11) is separated into two independent sets. The first one describes the $ppK^-$ system, while the second one corresponds to the $np\bar{K}^0$ system. Thus, the coupling between $ppK^-$ and $np\bar{K}^0$ is eliminated under the assumption $\alpha = \beta$.

One can obtain approximation for the set (10) taking $Y_2 = 0$ with the condition that the $p\bar{K}^0$ potential is weak and $U_1 \approx U_2$ and $W_1 \approx W_2$. The new set has the following form:

$$
\begin{aligned}
(H_0^U + v_{pp} - E)A_1 &= -v_{pp}(B_1 + Y_1) \ , \\
(H_0^W + v_{pK^-} - E)B_1 &= -v_{pK^-}(A_1 + Y_1) \ , \\
(H_0^Y + v_{pK^-} - E)Y_1 &= -v_{pK^-}(A_1 + B_1) \ , \\
(H_0^U + v_{pn} - E)A_2 &= -v_{pn}B_2 \ , \\
(H_0^W + v_{n\bar{K}^0} - E)B_2 &= -v_{n\bar{K}^0}A_2 \ .
\end{aligned}
\tag{12}
$$

The analysis of (12) shows that again we have two independent sets. The first one describes the $ppK^-$ system, while the second one corresponds to the $np\bar{K}^0$ system when the weak $p\bar{K}^0$ interaction is neglected. We will see below that the assumed conditions are satisfied in $ppK^-/np\bar{K}^0$ calculations. One can conclude that the parameters of the $pK^0$ interaction cannot be fixed by

the study of $ppK^-/np\bar{K}^0$ system due to small contribution of the corresponding $Y_2$ component. Note that the elimination $U_2 - U_1$ and $W_2 - W_1$ is not possible for the $nn\bar{K}^0/np K^-$ coupled system. In this case, the difference between $nn$ and $np$ spin triplet potentials violates the symmetry of the equations.

## 3 Numerical Results

The ground state energy of "$ppK^-$" cluster were calculated with effective AY potentials [12] for the $\bar{K}N$ interaction. We used the modified MT I-III potential [17] for the $NN$ nuclear interaction.

Firstly, let us pay attention to numerical analysis of Eq. (6) where we have single undefined constant $\alpha$ (or $\beta$). If $\alpha = 1$, $\beta = 0$ the system of equations (6) decouples and the first three equations describe the kaonic cluster $ppK^-$, while last three equations describe the kaonic cluster $pnK^0$. In opposite case we have coupling between the $ppK^-$ and $pn\bar{K}^0$ states. If the coupling, defined by the potentials $V_{12}$ and $V_{21}$, is ignored, the systems transit as one to another when the coupling constant $\alpha$ increases from 0 to 1. In Fig. 2 we present the results of calculations for the binding energies of the $ppK^-$ and $np\bar{K}^0$ versus $\alpha^2$ for the case when $V_{12}$ and $V_{21}$ are omitted in Eq. (6). Important is that the "$ppK^-$" cluster can be described as having two levels with the same energy for the $\alpha = \frac{1}{\sqrt{2}}$. Taking into account that the transition $pK^-/n\bar{K}^0$ makes the same probability for the $pK^-$ and $n\bar{K}^0$ states of $N\bar{K}$ system, we assume the same probability for $ppK^-$ and $np\bar{K}^0$ states. According to the two-level system theory, two levels of the $ppK^-/np\bar{K}^0$ system are crossing when the coupling is absent. When $\alpha^2 = \frac{1}{2}$ and $V_{12} = V_{21}=0$, the Hamiltonians corresponding to the $ppK^-$ and $np\bar{K}^0$ states are the same. It is initial point of the theory. Switching on the coupling, i.e. considering $V_{12}$ and $V_{21}$ as non-zero terms in Eq. (6), has to lead to "repulsion of the levels" or anti-crossing of the levels. Thus, the coupling constant has to be chosen as $\frac{1}{\sqrt{2}}$. It means that the probability of the $ppK^-$ and $np\bar{K}^0$ states is equal in the "$ppK^-$" cluster. The energy in this point is different from one obtained in the framework of the "traditional" isospin $NN\bar{K}$ model [16]. However, the same coupling constant was used in Ref. [13], where an isospin model was actually employed based on a "charge isospin basis". This choice for $\alpha$ creates a correspondence between the isospin and particle models.

The numerical results for the quasi-bound state energy $|E_{NN\bar{K}}|$ of the "$ppK^-$" cluster are presented in Table 1. We compare the results of the isospin and particle models. The result for isospin model is the same which was obtained early in Ref. [16] where the configuration space Faddeev equations was also used. It can be mentioned that the nucleon-nucleon interaction is appeared as an attractive one. The values for $|E_{NN\bar{K}}(V_{NN} = 0)|$ in the isospin model is smaller than the value of $|E_{NN\bar{K}}|$. In other words when the $NN$ interaction is switched off then the binding energy decreases. For the particle model we have calculated the energy in different cases. For the first one, we calculated energies taking into account the coupling between the $ppK^-$ and $np\bar{K}^0$ states. We obtained two energies $\epsilon_1$ and $\epsilon_2$, which are the lower and upper energies of the quasi-doublet state $ppK^-/np\bar{K}^0$. For the second one, we calculated energies of separated $ppK^-$ and $np\bar{K}^0$ states. In this case, we had also two values for $ppK^-$ and $np\bar{K}^0$ system, respectively.

The attraction/repulsion character of $pp$ potential at different distances can be explained by Table 1. The attraction/repulsion behavior of the $pp$ potential is appeared by comparison of the bound state energy to one when the interaction between two nucleons is turned off (case $V_{NN} = 0$). This effect is caused by strong attractive singlet $pK^-$ potential which allows the identical particles to be closer together. The compact system has larger binding energy. Within the isospin model, the energy is larger than in the case when the $NN$ potential is

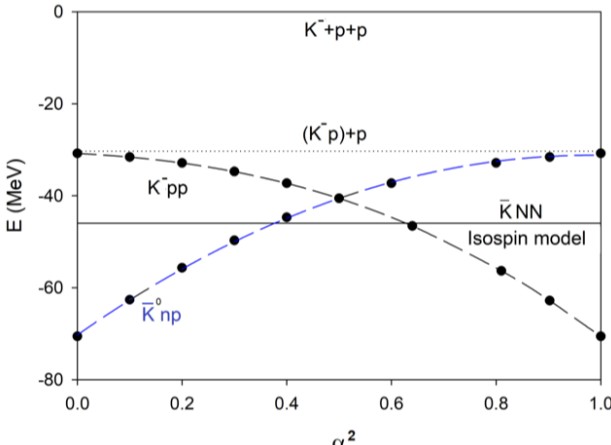

Figure 2: The binding energies of the $ppK^-$ and $np\bar{K}^0$ states as solutions of Eq. (6) when $V_{12} = 0$ and $V_{21} = 0$ for different values of the coupling parameter $\alpha$. The difference of the masses of kaons does not taken into account. The energies are measured from three-body threshold. The energy of two-body threshold is shown by the dotted line, while the solid line presents the energy of quasi-bound state obtained within the isospin model.

omitted. Thus, the $NN$ potential is weakly attractive. In the particle model, the $NN$ potential gives the repulsive effect due to the more compact spacial configuration of the system when the repulsive core of the potential becomes effective on short distances.

The numerical results for the $NN\bar{K}(s_{NN} = 1)$ system are presented in Table 2. The system is unbound within isospin consideration. Comparing the results to the results in Table. 1, we see that the $npK^-$ ($s_{NN} = 1$) level is located lower than $npK^-$ ($s_{NN} = 0$) on 7 MeV. It is effect of the difference of the states of nucleon pairs. The $s_{NN} = 1$ state (deuteron) corresponds to the bound state of the pair with energy of -2.23 MeV. This additional bound pair in the system $npK^-$ ($s_{NN} = 1$) leads to increasing three-body binding energy for the 7 MeV.

The nucleon-nucleon potential is not effective in triplet/triplet isospin/spin state. In used MT-I-III potential, the corresponding component is equal to zero. The results for the $nn\bar{K}^0$ and $ppK^-(V_{NN} = 0)$ systems are the same. We can evaluate the mass polarization effect [18] for $nn\bar{K}^0$ system as difference between $|E_{nn\bar{K}^0}|$ and $2|E_{N\bar{K}}|$. The value is about 20 MeV and the relative contribution of the mass polarization to the binding energy is about 25%. Notice, the last value depends mainly on the mass ratio of the particles in the system. For the system where the mass of non-identical particle is essentially larger then the mass of identical particles the effect can be neglected and $|E_3| \approx 2|E_2|$.

The results for the coupled $nn\bar{K}^0/npK^-$ system demonstrate so called "repulsion of levels", which we defined as extension of the energy distance between $nn\bar{K}^0$ and $npK^-$ levels due to the channel coupling. The upper level of the system with the energy $\epsilon_2$ becomes to be unbound. The lower level with the energy $\epsilon_1$ becomes to be more deeper.

It is interesting to mention that $nn\bar{K}^0$ system is significantly more bound than the $npK^-$ due to two strong $n\bar{K}^0$ interactions which are specularly associated with the Lambda(1405) state, while $npK^-$ is formed by the week $nK^-$ and strong $pK^-$ interactions. The attractive contribution of the spin triplet $np$ potential cannot compensate the weak contribution of the $nK^-$ potential.

Finally, we present spectrum of $ppK^-/np\bar{K}^0$ and $nn\bar{K}^0/npK^-$ bound and resonance states in Fig. 3. The effect of coupling is shown as a "repulsion of the levels". On the left hand side of Fig. 3, we present the results obtained for the separated systems $ppK^-$ and $np\bar{K}^0$ (also for

Table 1: The quasi-bound state energy $|E_{NN\bar{K}}|$ of the "$ppK^-$" cluster calculated within the isospin formalism and particle model with the AY $\bar{K}N$ and MT I-III nucleon-nucleon potentials. $|E_{N\bar{K}}|$ is the binding energy of the pair $N\bar{K}$ ($pK^-$ or $n\bar{K}^0$). The energies of the separated $ppK^-$ and $np\bar{K}^0$ systems are presented. The values of $|E_{NN\bar{K}}(V_{NN} = 0)|$ when the interaction between two nucleons is turned off are shown. $\epsilon_1$ and $\epsilon_2$ are the lower and upper energies of the quasi-doublet state $ppK^-/np\bar{K}^0$, respectively. The energies are given in MeV. The masses of the kaons (nucleons) are equal to the value for averaged mass.

| Model | System | $|E_{N\bar{K}}|$ | | $|E_{NN\bar{K}}|$ | $|E_{NN\bar{K}}(V_{NN} = 0)|$ |
|---|---|---|---|---|---|
| Isospin | $NN\bar{K}(s_{NN} = 0)$ | 30.3 | | 46.0 | 42.9 |
| Particle | $ppK^-/np\bar{K}^0$ | | $\epsilon_1$ | 70.6 | 80.8 |
| | | | $\epsilon_2$ | 30.3 | – |
| | $ppK^-$ | | | 70.6 | 80.8 |
| | $np\bar{K}^0$ | | | 30.8 | – |

Table 2: The quasi-bound state energy $|E_{NN\bar{K}}|$ of the $NN\bar{K}(s_{NN} = 1)$ system calculated within the isospin formalism and particle two-level system model with the AY $\bar{K}N$ and MT I-III nucleon-nucleon potentials. $|E_{N\bar{K}}$ is the binding energy of the singlet state of $N\bar{K}$ pair ($pK^-$ or $n\bar{K}^0$). The energies for the separated $nn\bar{K}^0$ and $npK^-$ systems are presented. The values of $|E_{NN\bar{K}}(V_{NN} = 0)|$ when the interaction between two nucleons is turned off are shown. $\epsilon_1$ and $\epsilon_2$ are the energies of the quasi-doublet of $nn\bar{K}^0/npK^-$ state, respectively. The energies are given in MeV.

| Model | System | $|E_{N\bar{K}}|$ | | $|E_{NN\bar{K}}|$ | $|E_{NN\bar{K}}(V_{NN} = 0)|$ |
|---|---|---|---|---|---|
| Isospin | $NN\bar{K}(s_{NN} = 1)$ | 30.3 | | unbound | – |
| Particle | $nn\bar{K}^0/npK^-$ | | $\epsilon_1$ | 87.7 | 80.8 |
| | | | $\epsilon_2$ | unbound | – |
| | $nn\bar{K}^0$ | | | 80.8 | 80.8 |
| | $npK^-$ | | | 37.9 | – |

separated $nn\bar{K}^0$ and $npK^-$). For comparison, the results for corresponding coupled system are shown on the right hand site. If the repulsion of levels is take place, the energy splitting for the quasi-doublets of coupled systems becomes larger that was for separated systems. One can see, that this effect is only appeared for the case $nn\bar{K}^0/npK^-$. According to Eq. (12), the effect is not visible for the $ppK^-/np\bar{K}^0$ case.

## 4 Conclusions

We proposed the two-level system treatment for $NN\bar{K}$ kaonic system based on particle representation. The cluster $NN\bar{K}(s_{NN} = 0)$ is presented as two-level system including $ppK^-$ and $pn\bar{K}^0$ states. The same approach is applied for the $NN\bar{K}(s_{NN} = 1)$ system considered as $nn\bar{K}^0/npK^-$ coupled states. The coupled coefficients were chosen to be $\alpha = \beta = \frac{1}{\sqrt{2}}$ for both $NN\bar{K}$ kaonic systems. It means that the probabilities to find the $NN\bar{K}(s_{NN} = 0)$ system as $ppK^-$ or $pn\bar{K}^0$ are equal.

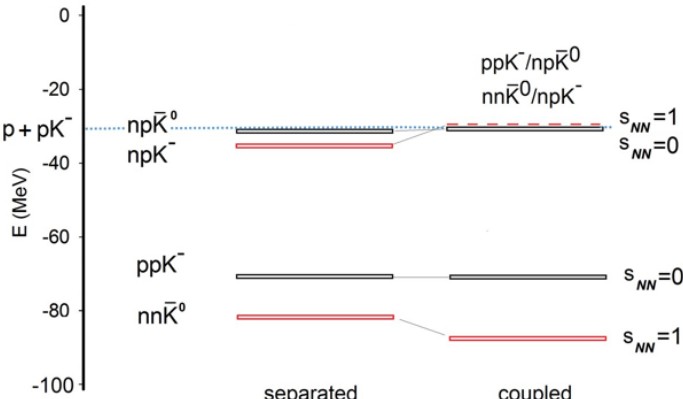

Figure 3: Spectrum of $ppK^-/np\bar{K}^0$ and $nn\bar{K}^0/npK^-$ bound (double lines) and resonance (dashed line) states. The two-body threshold $p+(pK^-)$ is shown by the dotted line. The effect of the coupling is presented as a "repulsion of levels" and is shown by connecting lines. The energies is reassured from the three-body threshold $p+p+K^-$. The spin of the nucleon pair is show.

We found deeply bound state of the $NN\bar{K}(s_{NN}=0)$ cluster with the energy about -72 MeV below $p+p+K^-$ threshold using the phenomenological AY and MT potentials. Also, there is weakly bound $pnK^-$ states with the energy -31 MeV. These states are corresponded to the separated channels $ppK^-$ and $np\bar{K}^0$, respectively. The $nn\bar{K}^0/npK^-$ system has deeply bound state with the energy above -87 MeV below $n+p+K^-$ threshold. This state corresponds to the $nn\bar{K}^0$ channel.

Thus, the sequential particle model for $NN\bar{K}$ kaonic cluster provides one deeply and one shallow bound states.

## Acknowledgments

This work is supported by the US National Science Foundation grant HRD-1345219.

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
