# Peer review of "Particle representation for $NN{\bar K}$ system"

_SciPost Physics Proceedings, doi:SciPost Phys. Proc. 3, 044 (2020)_

## Round 1 · Referee Report · Anonymous (Referee 1) · 2020-1-13

Report
The proceedings contribution "Particle representation for NN¯K system" by Vlahovic, Filikhin and Kezarashvili
provides a succinct summary of coupled-channels calculations in this system. It particularly focuses on the ppK- and npK0 channels, and provides an analytical discussion of how coupling between these two channels affects the theoretical understanding of the three-body problem (including extension of the Faddeev formalism to coupled channels). The paper has a clear introduction and justification, and is overall well written when it comes to general physics discussions.
While I think I understand the motivation of this work, I have trouble understanding some of the details. First, I think I see how the coupled channels approach can be introduced for the Faddeev formalism, but the physical role of the A-matrix in Eq. (2) is unclear to me. On the one hand, the mixing between different isospin states should be directly dictated by the laws of angular momentum/isospin (eg some Clebsch-Gordan coefficients for three, rather than 2, states). I presume similar decompositions exist for the case of three spins, for instance, and I would assume the same should hold for isospin systems.
The discussion around the freedom on the alfa/beta parameters, and the discussion of their impact, is given in Section 3 and in the context of Figure 2. Having done that, the authors immediately settle for \alpha=1/\sqrt{2}. I don't think I understand the motivation to choose this value (which in Fig 2 provides the same energies for the two systems). I suggest a more elaborate discussion is provided, and in particular I would also like to know whether the choice V_12=V_21=0 is fixed across the paper (or only at this stage).
Figure 3: the authors state in Table 2 that epsilon_2 is "unbound". If I understand correctly, epsilon_2 corresponds to the dashed line in the "coupled" panel (right), which indicates that it is bound (negative energy) but above the other state. Should the dashed line be above 0?
Finally, and in the context of the isospin splitting, is there a simple reason to see why the ppK or nnK systems are significantly more bound than the npK states? np pairs have a bound states but pp and nn don't, so I would have expected the kaon to produce further attraction in the np channel than in the nn/pp one. I don't think I found a justification of this behavior in the paper.
provides a succinct summary of coupled-channels calculations in this system. It particularly focuses on the ppK- and npK0 channels, and provides an analytical discussion of how coupling between these two channels affects the theoretical understanding of the three-body problem (including extension of the Faddeev formalism to coupled channels). The paper has a clear introduction and justification, and is overall well written when it comes to general physics discussions.
While I think I understand the motivation of this work, I have trouble understanding some of the details. First, I think I see how the coupled channels approach can be introduced for the Faddeev formalism, but the physical role of the A-matrix in Eq. (2) is unclear to me. On the one hand, the mixing between different isospin states should be directly dictated by the laws of angular momentum/isospin (eg some Clebsch-Gordan coefficients for three, rather than 2, states). I presume similar decompositions exist for the case of three spins, for instance, and I would assume the same should hold for isospin systems.
The discussion around the freedom on the alfa/beta parameters, and the discussion of their impact, is given in Section 3 and in the context of Figure 2. Having done that, the authors immediately settle for \alpha=1/\sqrt{2}. I don't think I understand the motivation to choose this value (which in Fig 2 provides the same energies for the two systems). I suggest a more elaborate discussion is provided, and in particular I would also like to know whether the choice V_12=V_21=0 is fixed across the paper (or only at this stage).
Figure 3: the authors state in Table 2 that epsilon_2 is "unbound". If I understand correctly, epsilon_2 corresponds to the dashed line in the "coupled" panel (right), which indicates that it is bound (negative energy) but above the other state. Should the dashed line be above 0?
Finally, and in the context of the isospin splitting, is there a simple reason to see why the ppK or nnK systems are significantly more bound than the npK states? np pairs have a bound states but pp and nn don't, so I would have expected the kaon to produce further attraction in the np channel than in the nn/pp one. I don't think I found a justification of this behavior in the paper.

Author: Igor Filikhin on 2020-01-24 [id 718]
(in reply to Report 1 on 2020-01-13)Dear Referee,
We thank you for carefully reading our manuscript and making insightful comments that will enhance the clarity of our work. We greatly value your time and energy in helping us improve our manuscript. We are humbled by your assessment that our fellow researchers will find our work relevant and interesting. We completely agree with your comments and have addressed your concerns individually below.
Sincerely,
B. Vlahovic, I. Filikhin, R. Ya. Kezarashvili
Reviewer
The proceedings contribution "Particle representation for NN¯K system" by Vlahovic, Filikhin and Kezarashvili
provides a succinct summary of coupled-channels calculations in this system. It particularly focuses on the ppK- and npK0 channels, and provides an analytical discussion of how coupling between these two channels affects the theoretical understanding of the three-body problem (including extension of the Faddeev formalism to coupled channels). The paper has a clear introduction and justification, and is overall well written when it comes to general physics discussions.
Our Response
Thank you for the favorable assessment of our manuscript.
Reviewer
While I think I understand the motivation of this work, I have trouble understanding some of the details. First, I think I see how the coupled channels approach can be introduced for the Faddeev formalism, but the physical role of the A-matrix in Eq. (2) is unclear to me. On the one hand, the mixing between different isospin states should be directly dictated by the laws of angular momentum/isospin (eg some Clebsch-Gordan coefficients for three, rather than 2, states). I presume similar decompositions exist for the case of three spins, for instance, and I would assume the same should hold for isospin systems.
Our Response
Thank you very much for this comment. We added after Eq. (5), the second line, the following explanation in the revised manuscript:
Thus, the matrix $A$ describes the coupling $ppK^{-}/np{\bar{K}}^{0}$ via non-diagonal elements $V_{12}$ and $V_{21}$.
The isospin states of $ppK^{-}$ and $np{\bar{K}}^{0}$ are the same and the possible transformation $pK^{-}$ $\rightarrow$ $n{\bar{K}}^{0}$ can be described as a coupling
between the systems.
Reviewer
The discussion around the freedom on the alfa/beta parameters, and the discussion of their impact, is given in Section 3 and in the context of Figure 2. Having done that, the authors immediately settle for \alpha=1/\sqrt{2}. I don't think I understand the motivation to choose this value (which in Fig 2 provides the same energies for the two systems). I suggest a more elaborate discussion is provided, and in particular I would also like to know whether the choice V_12=V_21=0 is fixed across the paper (or only at this stage).
Our Response
Thank you for this this insightful and interesting comment. The choice V_12=V_21=0 is not fixed across the paper, it is only at this stage for the results shown in Figure 2. For more clarity we wrote (page 6, second paragraph, line 5):
If the coupling, defined by the potentials $V_{12}$ and $V_{21}$, is ignored, the systems transit as one to another when the coupling
constant $\alpha$ increases from 0 to 1. In Fig. \ref{fig3a} we present the results of calculations for the binding energies of the $ppK^-$ and $np\bar K^0$ versus
$\alpha^2$ for the case when $V_{12}$ and $V_{21}$ are omitted in Eq. (\ref{eq:11c}).
Important is that the "$ppK^-$" cluster can be described as having two levels with the same energy for the $\alpha=\frac{1}{\sqrt{2}}$.
Taking into account that the transition $pK^-$/$n\bar K^0$ makes the same probability for the $pK^-$ and $n\bar K^0$ states of $N\bar K$ system, we assume the same probability for $ppK^-$ and $np\bar K^0$ states. According the two-level system theory, two levels of the $ppK^-$/$np\bar K^0$ system
are crossing when the coupling is absent. When $\alpha^2=\frac12$ and $V_{12}=V_{21}$=0, the Hamiltonians corresponding to the $ppK^-$ and $np\bar K^0$ states are the same. It is initial point of the theory.
Switching on the coupling, i.e. considering $V_{12}$ and $V_{21}$ as non-zero terms in Eq. (\ref{eq:11c}), has to lead to "repulsion of the levels" or anti-crossing of the levels.
Thus, the coupling constant has to be chosen as $\frac{1}{\sqrt{2}}$. It means that the probability of the $ppK^-$ and $np{\bar K}^0$ states is equal in the "$ppK^-$" cluster. The energy in this point is different from one obtained in the framework of the "traditional" isospin $NN\bar K$ model \cite{Kezerashvili2016}. However, the same coupling constant was used in Ref. \cite{R16}, where an isospin model was actually employed based on a "charge isospin basis". This choice for $\alpha$ creates a correspondence between the isospin and particle models.
Reviewer
Figure 3: the authors state in Table 2 that epsilon_2 is "unbound". If I understand correctly, epsilon_2 corresponds to the dashed line in the "coupled" panel (right), which indicates that it is bound (negative energy) but above the other state. Should the dashed line be above 0?
Our Response
Many thanks for this comment. The dashed line is above the two-body threshold, not shown in the figure, which corresponds the breakup of the system on two fragments and it is in the negative region of energies. In the revised manuscript we replace Figure 3 by the new one, where explicitly is shown the two-body threshold p+(pK^-) by the dotted line., as well as we added in the caption for Figure 3:
The two-body threshold $p+(pK^-)$ is shown by the dotted line.
Reviewer
Finally, and in the context of the isospin splitting, is there a simple reason to see why the ppK or nnK systems are significantly more bound than the npK states? np pairs have a bound states but pp and nn don't, so I would have expected the kaon to produce further attraction in the np channel than in the nn/pp one. I don't think I found a justification of this behavior in the paper.
Our Response
Thank you for this this insightful and interesting comment. For the clarification of this situation we added the following text (page 7, last paragraph):
It is interesting to mention that $nn\bar K^0$ system is significantly more bound than the $npK^-$ due to two strong $n\bar K^0$ interactions which are specularly associated with the Lambda(1405) state, while $npK^-$ is formed by the week $nK^-$ and strong $pK^-$ interactions. The attractive contribution of the spin triplet $np$ potential %(adding the energy to $NN$) pair
cannot compensate the weak contribution of the $nK^-$ potential.
Attachment:
RK_SciPost_particle_rep_VFK_081.pdf

---

## Editorial Decision

published